# Downlink Power Allocation Strategy for Next-Generation Underwater Acoustic Communications Networks

**Ishtiaq Ahmad [1,2]** and **KyungHi Chang [1,\*]**

[1] Department of Electronic Engineering, Inha University, Incheon 22212, Korea; ishtiaqahmad@gu.edu.pk or ishtiaq001@gmail.com

[2] Department of Electrical Engineering, Gomal University, Dera Ismail Khan 29200, Pakistan

\* Correspondence: khchang@inha.ac.kr

**Abstract:** The increasing interest in next-generation underwater acoustic communications networks is due to vast investigation of oceans for oceanography, commercial operations in maritime areas, military surveillance, and more. A surface buoy or underwater base station controller (UBSC) communicates with either transceivers or underwater base stations (UBSs) via acoustic links. Transceivers further communicate with underwater sensor nodes using acoustic links. In this paper, we employ a downlink (DL) power allocation (PA) strategy using an orthogonal frequency-division multiple access (OFDMA) technique for underwater acoustic communications (UAC) networks. First, we present an approach to power offsets using three kinds of pilot spacing and apply the power boosting (PB) concept on orthogonal frequency-division multiplexing (OFDM) symbols for the UAC network. Secondly, we draw the block error rate (BLER) curves from link-level simulation (LLS) and analyze the signal-to-noise ratio (SNR) for both PA and non-PA strategies. Lastly, we adopt the best PB for system-level simulation (SLS) and compare the throughput and outage performance for PA and non-PA strategies. Hence, the simulation results confirm the effectiveness of the DL PA strategy for UAC networks.

**Keywords:** UAC network; underwater channel model; power allocation; LLS; SLS

## 1. Introduction

Next-generation underwater acoustic communications (UAC) networks have the capability to observe and explore the aquatic environment. Therefore, UAC networks are widely considered for long-distance underwater communications. Quality of service (QoS) requirements are needed in order to fill military and scientific data collections about the ocean floor. However, it is difficult to manage UAC channel properties, such as delay and Doppler spread, which cause severe fading [1–5].

Compared with the terrestrial channel model, the UAC channel model poses difficult challenges. In addition, bandwidth availability is the most prominent challenge in UAC networks. Therefore, cellular and frequency-reuse concepts are more tempting when trying to improve the coverage and capacity of UAC networks [6–8]. In this paper, we consider a cellular type of UAC network architecture and employ a downlink (DL) power allocation (PA) strategy to analyze system throughput and outage performance in a system-level simulation (SLS). Evaluation of a link-level simulation (LLS) is done by measuring the signal-to-noise ratio (SNR) versus block error rate (BLER) [9–13].

Research has been proposed to assess the features of UAC networks in the existing literature. However, features related to UAC networks still need to be addressed on an emergent basis, including consideration of complicated scenarios such as terrestrial cellular networks [14–27]. The existing works

related to underwater communication have been considered simple network architectures and have been mostly focus on the assessment of underwater channel model and routing protocols [28–31]. In order to fill this gap, we consider the complicated scenarios associated with terrestrial networks and employ the downlink power allocation strategy for the UAC networks. Many researchers who have previously worked on UAC power allocation issues have not considered the associated complicated scenarios.

### 1.1. Differences in System Methodologies in the Literature

In a study by Cheon and Cho (2017) [32], an equal transmission power control scheme was applied to the clustered based network approach for UAC networks. The major difference in this study is that we employ the power allocation scheme for non-orthogonal multiple access while utilizing the orthogonal frequency-division multiplexing (OFDM) technique for UAC networks. In [33], the authors investigated the power allocation strategy for energy harvesting in UAC networks. They considered two scenarios for knowing the channel state information and applied stochastic dynamic programming to find the optimal power allocation for UAC networks. The major difference in this study is that we adopt the power allocation for energy harvesting in the UAC network. In [34], the authors jointly utilized the power and frequency allocation strategy to minimize the energy consumption of UAC networks. The major difference in this study is that we select the proper center frequency, bandwidth, and transmission using the routing protocols. Hence, the different approaches and system design in the existing works [32–34] resulted in different system parameters. Therefore, it is very difficult to compare the proposed DL PA strategy with other research. To the best of our knowledge, this work is the first to present the downlink power allocation issues using the power allocation strategy for UAC networks.

### 1.2. Main Contributions

The main contributions of this paper are as follows:

- The DL PA strategy is employed using an orthogonal frequency-division multiple access (OFDMA) technique for UAC networks.
- The power offset approach is presented using three kinds of pilot spacing and by applying the power boosting (PB) concept on OFDM symbols for a UAC network.
- BLER results are drawn from the LLS, and we analyze the SNR for PA and non-PA strategies.
- The best PB case is adopted for the SLS, and we compare throughput and outage performance for PA and non-PA strategies.

The rest of the paper is organized as follows. In Section 2, we provide the system model for UAC networks. In Section 3, we discuss the proposed DL PA strategy in detail for UAC networks. In Section 4, the performance of the proposed DL PA strategy is assessed by using the LLS and SLS results from the UAC network. Finally, we conclude the paper in Section 5.

## 2. System Model for UAC Networks

We built MATLAB-based LLS and SLS platforms and employed the DL power allocation strategy for the next-generation UAC networks by referring the terrestrial cellular network communication approaches [6–8]. This work is a continuation of our previous work in [15] where we analyzed effective SNR mapping and link adaptation strategies for UAC networks. Therefore, we did not utilize common network simulators, for example DESERT under NS2.

### 2.1. Network Layout

We adopted the cellular concept in this paper [14–21], which is based on one-tier cellular structure, as shown in Figure 1. The red circles and green squares represent the underwater base station controllers (UBSCs) and underwater base stations (UBSs), respectively. The UBSCs were separated from each other

based on a 40 km intersite distance. The center cell was the region of interest, which is highlighted in yellow in Figure 1. Three UBSs were connected to the UBSC via acoustic communications linked with fixed distances, i.e., short (1 km), medium (5 km), and long (10 km) [15]. The scenario of acoustic communication between the UBSC and UBS was quite similar to terrestrial cellular communication, such as base stations and users, respectively. Therefore, the UBSs, which existed in the region of interest, could be considered to be the only users or receivers where the scheduling, outage, and throughput calculations were performed while the transmitters and receivers in first tier could be considered to be the interference providing nodes. Hence, the downlink power allocation strategy was implemented based on the scenario in Figure 1.

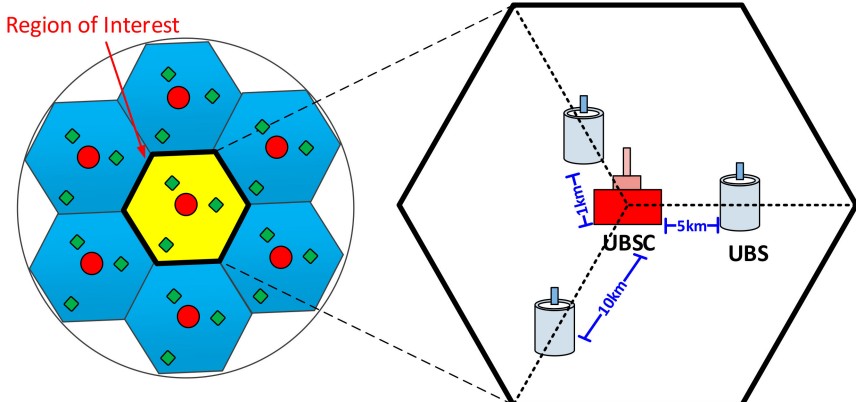

**Figure 1.** Cellular layout for an underwater acoustic communications (UAC) network. UBSC: underwater base station controllers; UBS: underwater base stations.

*2.2. Channel Model*

***Transmission Loss and Ambient Noise:*** The core difference between a terrestrial network and an underwater network is the underwater channel model. We chose the most representative transmission loss model, which has been used in a lot of previous research [3,29–31].

***Fading Channel Model:*** We utilized the twelve-path Rician channel in the south sea by conducting an experiment for data acquisition from 9 December 2017 to 13 December 2017. The experiment was conducted in Korea's Pohang Sea, Geoje City, Gyeongsangnamdo [15]. We drew the channel impulse response (CIR) of the south sea channel, as shown in Figure 2. There were a total of 54 symbols in one frame. We drew discrete and continuous CIRs for different frames, as shown in Figure 2a,b, respectively.

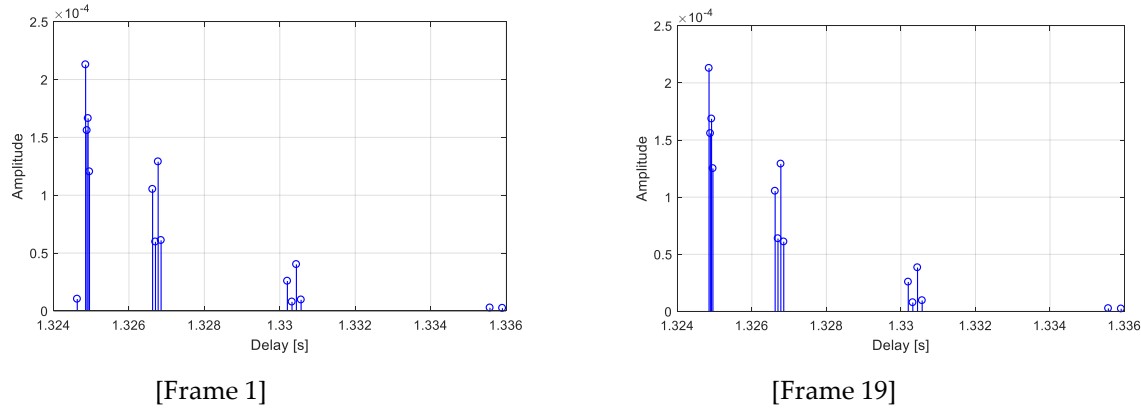

[Frame 1]                                    [Frame 19]

**Figure 2.** *Cont.*

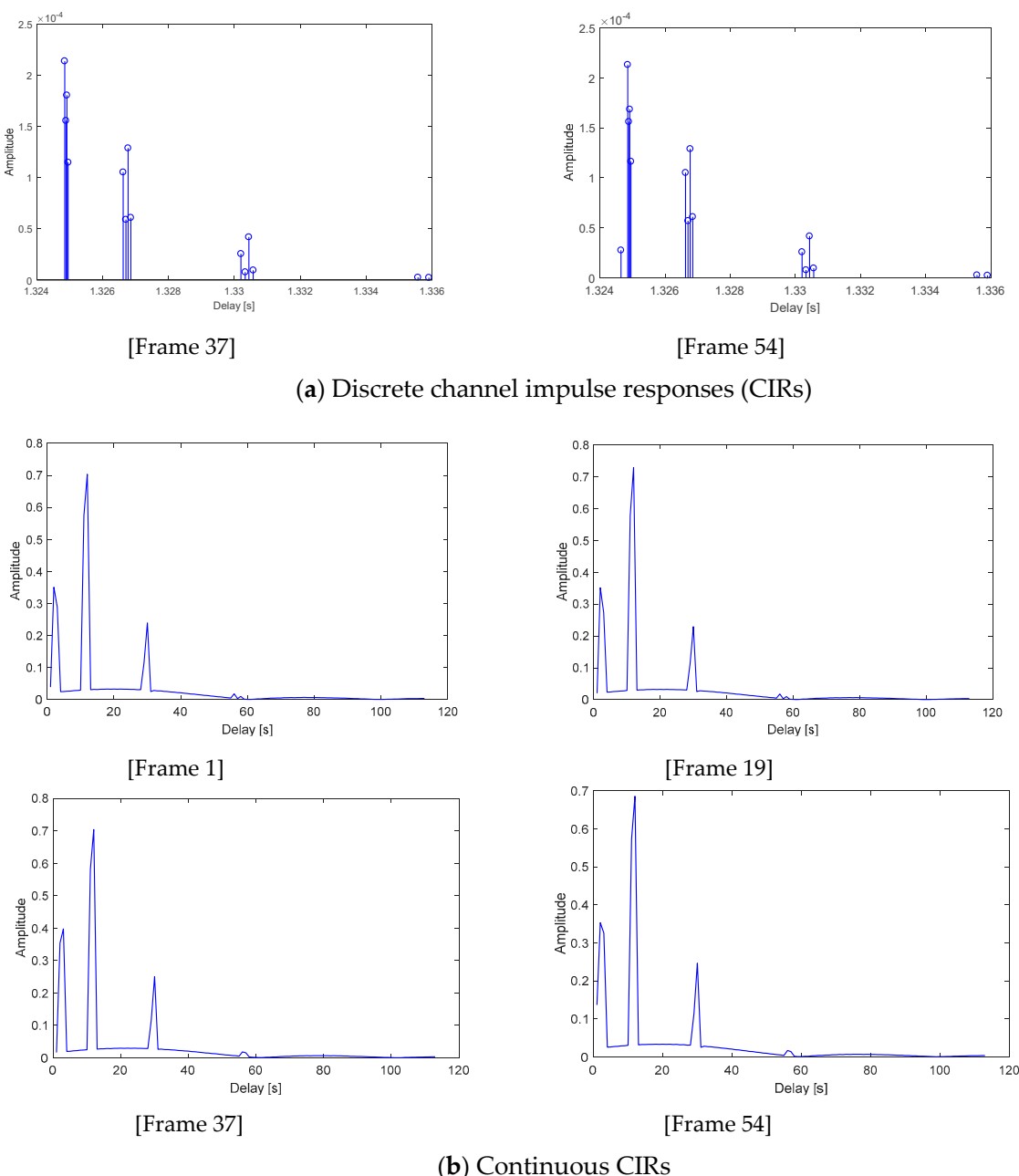

(**a**) Discrete channel impulse responses (CIRs)

(**b**) Continuous CIRs

**Figure 2.** Discrete and continuous CIRs for different frames: (**a**) discrete CIRs; (**b**) continuous CIRs.

Figure 2 shows that the behavior of the UAC south sea channel is quite similar to the statistical channel models in terrestrial scenarios (e.g., Veh. A, Veh. B, Winner II, etc.). The example behavior of terrestrial Winner II (scenario B1) channel model is shown in Figure 3 [35]. Hence, the delay versus the amplitude of discrete curves shows similar behavior for UAC south fading channel model and terrestrial Winner II fading channel model in Figures 2 and 3, respectively.

At LLS, we draw the BLER curves by considering the useful modulation and coding scheme levels [15], as shown in Figure 4. The corresponding look-up table is listed in Table 1. We took the SNR values at 10% of the BLER curves for the nine modulation and coding scheme levels and utilized these values in SLS for assessing the DL power allocation strategy.

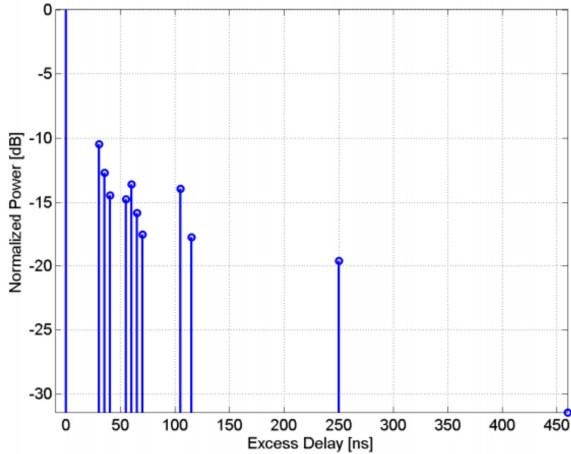

**Figure 3.** Excess delay versus normalized power of terrestrial winner II (B1 scenario) channel model.

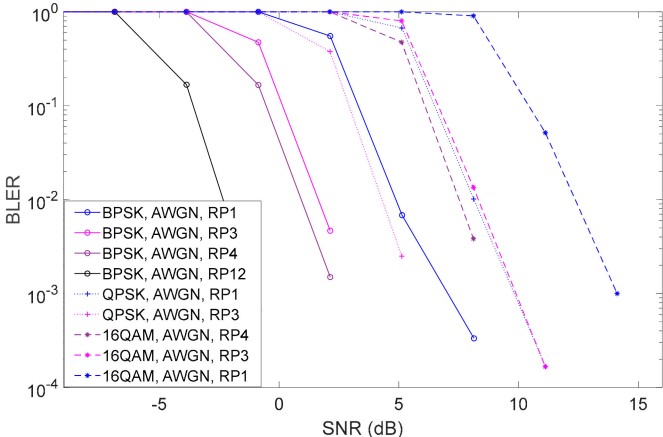

**Figure 4.** Final MCS (Modulation and Coding Scheme) levels.

**Table 1.** Channel quality indicator (CQI) Table for MCS levels of adaptive modulation in a UAC network system-level simulation (SLS).

| CQI | Modulations with Repetition Patterns | Code Rate | SNR (dB) |
|-----|--------------------------------------|-----------|----------|
| 1 | BPSK, RP12 | 1/2 | −3.6 |
| 2 | BPSK, RP4 | 1/2 | −0.6 |
| 3 | BPSK, RP3 | 1/2 | 0.1 |
| 4 | QPSK, RP3 | 1/2 | 2.9 |
| 5 | BPSK, RP1 | 1/2 | 3.4 |
| 6 | 16QAM, RP4 | 1/2 | 6.1 |
| 7 | QPSK, RP1 | 1/2 | 6.9 |
| 8 | 16QAM, RP3 | 1/2 | 7.2 |
| 9 | 16QAM, RP1 | 1/2 | 10.4 |

## 3. Proposed Downlink Power Allocation Strategy for UAC Networks

In this section, we explain the proposed DL PA strategy for the UAC network by referring to the 3rd generation partnership project (3GPP) long-term evolution (LTE) DL PA strategy [36–38]. There is no DL PA strategy available in UAC specification documents similar to 3GPP LTE; therefore, it is necessary to refer the already well-established DL PA approaches of a LTE system for UAC networks. The 3GPP has defined two parameters $P_A$ and $P_B$ that control downlink power allocation. $P_A$ represents the power of the reference signal when compared with OFDM symbol A and $P_B$ represents the power of the reference elements of OFDM symbol B as compared with OFDM symbol A. $P_A$ and $P_B$ have

various combinations of different values, which can be used to utilize the maximum power of the network, as per our requirements.

The main objective was to avoid power variations at the receiving side by maintaining constant power levels by using the DL PA strategy. The boost in the reference signal power is caused by the compensation of type B symbols (which contain reference symbols) as compared with the type A symbols (that do not contain reference symbols). The data signal power always depends on the number of allocated resource blocks. This allocation of resource blocks can be changed subframe by subframe. It is ensured that the overall OFDM symbol power should remain constant during the incorporation of the $P_A$ and $P_B$.

### 3.1. Power Offset Using Three Types of Pilot Spacing

From Figure 5, we can see that three kinds of pilot spacing are considered in which the power offset can be employed. The type A symbols ($P_A$) represent power offset between the reference signal (RS) power and the data signal power when there is no reference signal in the OFDM symbol. Because this symbol does not include the RS, it allocates more power to the data signal, keeping the same power per symbol. On the other hand, the type B symbols ($P_B$) are used to represent the power offset of OFDM symbols that contain reference symbols. We can allocate the power symbol by symbol. There are some symbols with a reference signal, and there are other symbols without a reference signal. We can put less power into the non–reference-signal channels with the symbol-carrying reference signal. According to Figure 5, there are four power offsets, $P_B$ 0, $P_B$ 1, $P_B$ 2, and $P_B$ 3, corresponding to pilot spacing [0, 0] [6, 3], [4, 2], and [2, 1], respectively. It should be noted that the $P_B$ 0 power offset for pilot spacing [0, 0] is equivalent to a non-PA strategy. Therefore, only the three PA cases based on pilot spacing [6, 3], [4, 2], and [2, 1] are considered.

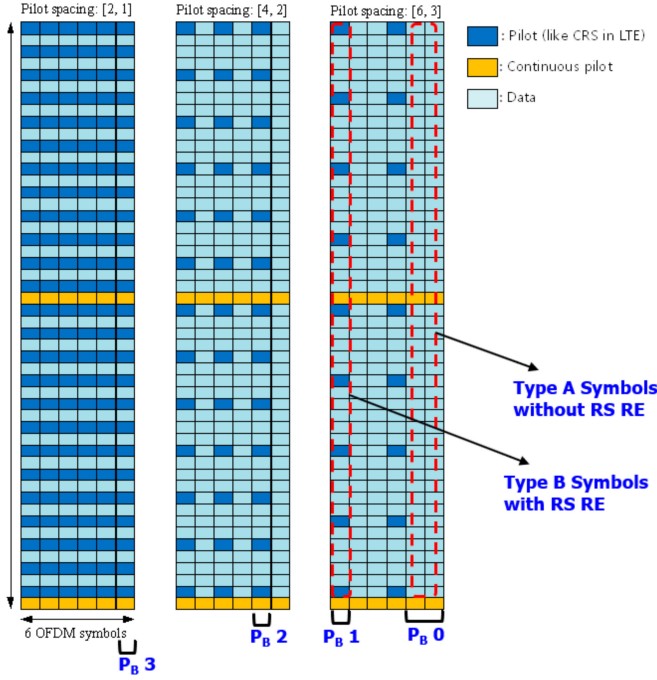

**Figure 5.** Power offsets based on pilot spacing [6, 3], [4, 2], and [2, 1].

### 3.2. Reference Signal Power Boosting Gain

The UBSC broadcasts the power levels of the cell-specific reference signal (CRS) and the data signals using reference signal power parameters $P_A$ and $P_B$. Transmit power of a resource element (RE) carrying the RS (in decibel-milliwatts) is specified as reference signal power. $P_A$ influences the parameter $\rho_A$, which is the ratio of the transmit power of the data signal (DS) RE to the transmit power of the RS RE; $\rho_A$ is applicable to OFDM symbols that do not carry an RS. $P_B$ establishes the relationship

between $\rho_A$ and $\rho_B$, where $\rho_B$ is the ratio of the transmit power of DS RE to the transmit power of the RS RE in the OFDM symbols that carry an RS.

Power boosting is mainly performed on the RS. However, because radio power is shared equally by all REs, the power allocation for each RE is fixed. By increasing the number of REs being used as a reference signal, the RS can be "boosted" by $2 \times$ (3 dB), $3 \times$ (4.7 dB), or $4 \times$ (6 dB) accordingly. The power boosting value is 0 if there are no extra resources used for the RS. $P_B = 0$: there is no increment on the CRS RE (that is, no power boosting), and the DS power is 1. $P_B = 1$: the RS power increases on the CRS RE 2 units (that is, power boosting), and the DS power on symbol type B is reduced to 4/5. $P_B = 2$: the RS power increases on the CRS RE 3 units (that is, power boosting), and the DS power on symbol type B is reduced to 3/5. $P_B = 3$: the RS power increases on the CRS RE 4 units (that is, power boosting), and the DS power on symbol type B is reduced to 2/5. A power boosting example is shown in Figure 6.

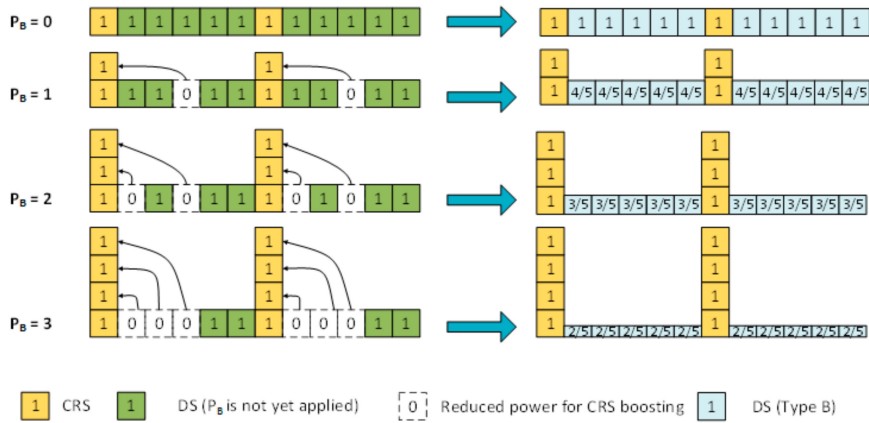

**Figure 6.** Power boosting for type B orthogonal frequency-division multiplexing (OFDM) symbols.

## 4. Performance Evaluation of the Proposed Downlink Power Allocation Strategy

The performance of the proposed DL PA strategy is assessed in this section for UAC networks. On the LLS side, we analyzed the BLER versus the SNR for PA and non-PA strategies using the south sea fading channel model. This provided the best PB case selection for SLS. Hence, we can assess the throughput and outage performance using the best PB case in a SLS under the one-tier UAC network shown in Figure 1. The main simulation parameters are listed in Table 2.

**Table 2.** System-level simulation parameters.

| Parameters | Values |
|---|---|
| Carrier Frequency | 5.5 kHz |
| Bandwidth | 5 kHz |
| No. of UBSCs | 1-tier, 7 Sites |
| Inter-site Distance | 4 km |
| No. of UBSs | 21 |
| Distances of UBSs from the UBSC | 10 km, 5 km, 1 km (long, medium, short distances) |
| Antenna Pattern Transmission Power | Omni-directional 46.989 dBm |
| Channel Model | Ambient Noise Pathloss Fading Channel Model |
| Transmission Modes | SISO (Single Input Single Output) |
| Effective SINR (Signal to Interference and Noise Ratio) | EESM (Exponential Effective SINR Mapping) |
| Scheduling | Proportional Fair |

*4.1. Analysis of PA and Non-PA Using LLS Results*

We considered three power allocation cases ($P_B$ 1, $P_B$ 2, and $P_B$ 3) corresponding to power offsets on the three kinds of pilot spacing shown in Figure 5. So, the power boosting on the CRS is 3 dB, 4.7 dB, and 6 dB for $P_B$ 1, $P_B$ 2, and $P_B$ 3, respectively. We drew the BLER versus SNR curves for the three modulation schemes, i.e., BPSK, QPSK, and quadrature amplitude modulation 16 (16QAM), using data repetition pattern (RP) 1 under the south sea fading channel model. Figure 7 shows the BLER curves for the PA and non-PA strategies. Figure 7a shows the non-PA strategy for pilot spacing [0, 0], which means that no PB is applied to the CRS. Figure 7b–d shows the PA strategies based on PB on the CRS at 3 dB, 4.7 dB, and 6 dB, respectively. Using Figure 7, the SNR comparison of MCS levels at $10^{-1}$ BLER under the south sea fading channel is in Table 3. We compared the SNR using the following Equation:

$$SNR = (\text{Average SNR from non-PA}) - (\text{Average SNR from PA}) \tag{1}$$

**Table 3.** Signal-to-noise ratio (SNR) comparison of MCS levels at $10^{-1}$ block error rate (BLER) on a south sea fading channel.

| MCS (Modulation and Coding Scheme) | CR (Code Rate) | PB (Power Boosting): 0 dB | PB: 3 dB | PB: 4.7 dB | PB: 6 dB |
|---|---|---|---|---|---|
| BPSK, RP1 | 1/2 | 13.72 | 12.62 | 13.06 | 12.7 |
| QPSK, RP1 | 1/2 | 16.15 | 15.31 | 14.83 | 15 |
| 16QAM, RP1 | 1/2 | 22.72 | 21.8 | 21.75 | 22.09 |
| Average SNR at BLER $10^{-1}$ | 1/2 | 17.53 | 16.57 | 16.54 | 16.59 |

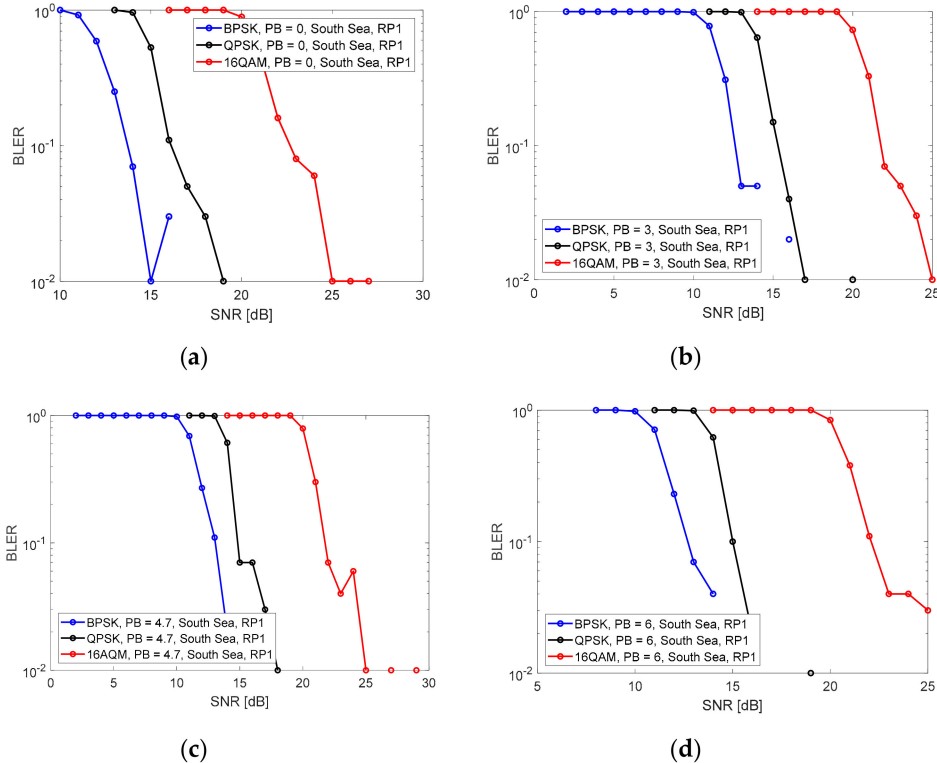

**Figure 7.** (**a**) Non-power allocation (PA) [0 dB], (**b**) PA [3 dB], (**c**) PA [4.7 dB], and (**d**) PA [6 dB].

Table 3 provides the SNR comparison for non-PA (PB: 0 dB) versus PA (PB: 3 dB, PB: 4.7 dB, and PB: 6 dB). We inserted the values from Equation (1) using PA and non-PA according to Table 3. The SNR for PB [3 dB], PB [4.7 dB], and PB [4.7 dB] is 0.96 dB, 0.99 dB, and 0.94 dB, respectively. According to

Equation (1), PA PB [4.7 dB] had a higher SNR; therefore, PA for $P_B$ 2 showed the best performance among the PA strategies. Hence, we adopted PA on $P_B$ 2 for the UAC network.

We utilized the SNR at 10% of the BLER LLS results using the EESM (Exponential effective SINR mapping) method for link-to-system mapping in the UAC network. Figures 8 and 9 show the BLER results based on non-PA, and with the best PA strategy ($P_B$ 2), respectively. The SNR versus the channel quality indicator (CQI) mapping based on the nine MCS levels for adaptive modulation in the UAC network SLS is shown in Table 4.

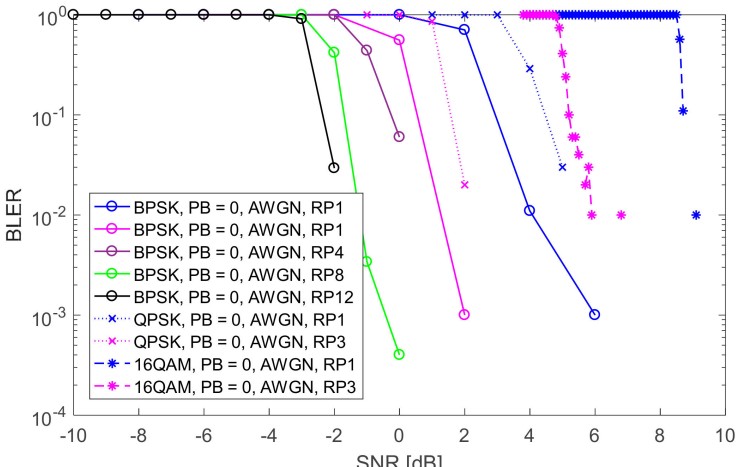

**Figure 8.** BLER curves for $P_B$ 0 [same scenario as non-PA].

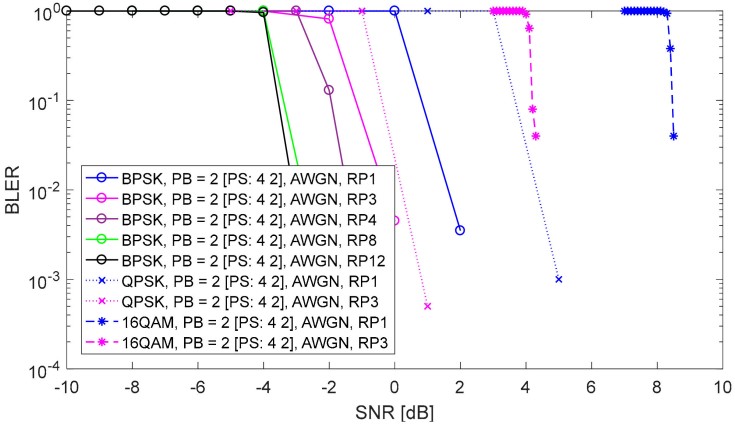

**Figure 9.** BLER curves for PA scenario $P_B$ 2.

**Table 4.** SNR comparison of MCS levels at $10^{-1}$ BLER under the south sea fading channel.

| CQI | MCS | CR | Case 0 [PB: 0 dB SNR dB] | Case 2 [PB: 3 dB SNR dB] |
|---|---|---|---|---|
| 1 | BPSK, RP12 | 1/2 | −2.4 | −3.6 |
| 2 | BPSK, RP8 | 1/2 | −1.8 | −3.4 |
| 3 | BPSK, RP4 | 1/2 | −0.2 | −1.99 |
| 4 | BPSK, RP3 | 1/2 | 0.75 | −1.2 |
| 5 | QPSK, RP3 | 1/2 | 1.7 | −0.4 |
| 6 | BPSK, RP1 | 1/2 | 2.95 | 0.9 |
| 7 | QPSK, RP1 | 1/2 | 4.5 | 3.8 |
| 8 | 16QAM, RP3 | 1/2 | 5.6 | 4.2 |
| 9 | 16QAM, RP1 | 1/2 | 8.9 | 8.2 |

### 4.2. Analysis of PA and Non-PA Using SLS Results

To verify the effectiveness of the proposed power allocation strategy, system-level simulations were performed with a one-tier UAC network. Our main target was to prove the proposed PA allocation strategy works efficiently in the UAC network. From Table 3, we can see that PA strategy 2 [P$_B$ 2] provided the best performance. Therefore, we utilized the MCS LLS results for PA strategy 2 using the EESM approach to analyze the throughput and outage performance.

Figure 10 shows that throughput was higher when using the south sea fading channel model. On the other hand, the west sea fading channel had severe fading effects that yielded less throughput than the Rician and south sea fading channels. Moreover, we compared the throughput at the 50th percentile of the cumulative distribution function (CDF) based on PA and non-PA.

From Table 5, PA strategy 2 (P2) is better than strategy 0 (P0), because the throughput increased when we applied the power allocation concept in a UAC .

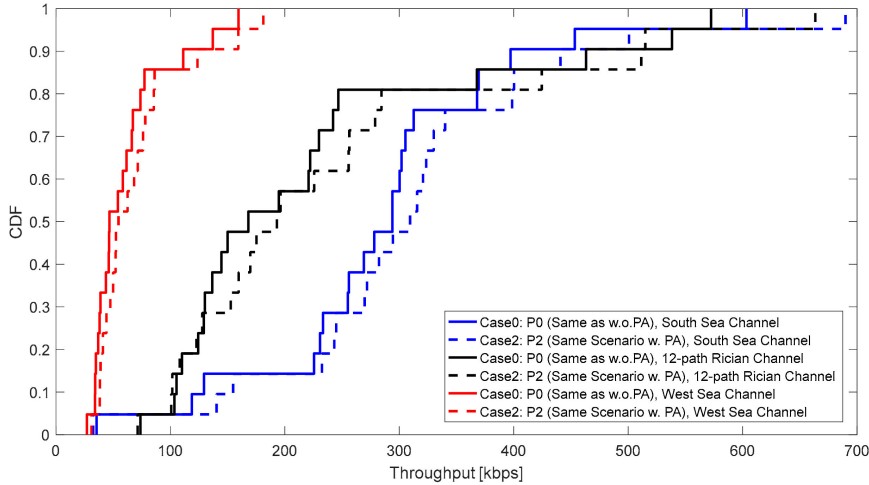

**Figure 10.** Throughput performance comparison for PA case 0: P0 versus PA case 1: P2.

**Table 5.** Throughput comparison at 50% cumulative distribution function (CDF).

| Fading Channel Models | Case 0 [P0 (kbps)] | Case 2 [P2 (kbps)] |
|:---:|:---:|:---:|
| South Sea | 293.9 | 309.4 |
| twelve-path Rician | 168 | 193 |
| West Sea | 46.7 | 54.6 |

In Figure 11, the outage performance was compared for PA strategy 0 (P0) versus PA strategy 1 (P2). Similarly, the south sea fading channel had better outage probability performance than the Rician and the west sea fading channels. As shown in Table 6, PA strategy 2 (P2) was better than strategy 0 (P0), because the outage probability decreased when power allocation was applied in the UWN.

Hence, from the above SLS analysis on throughput and outage performance, we confirmed that the proposed DL PA strategy works effectively in a UAC network.

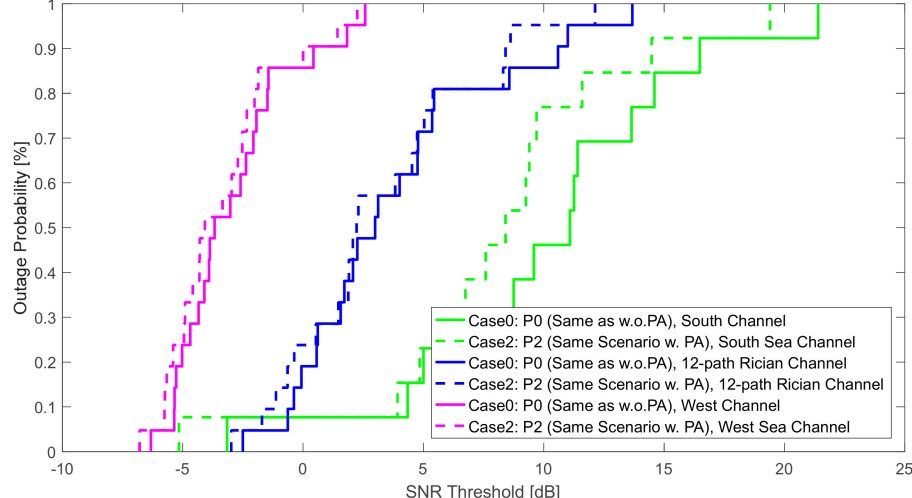

**Figure 11.** Outage performance comparison for PA case 0: P0 versus PA case 1: P2.

**Table 6.** Outage comparison at 50% CDF.

| South Sea Channel (At 5 dB SNR Threshold) | | 12-path Rician Channel (At 0 dB SNR Threshold) | | West Sea Channel (At −5 dB SNR Threshold) | |
|---|---|---|---|---|---|
| P0 | P2 | P0 | P2 | P0 | P2 |
| 24% | 16% | 24% | 21% | 33% | 24% |

## 5. Conclusions

This paper analyzed a downlink power allocation strategy for next-generation UAC networks. To the best of the authors' knowledge, this is the first study to explore power allocation issues in UAC networks. In this paper, we introduced the power offset concept based on three kinds of pilot spacing for UAC networks. Power boosting was employed on the CRS, and we drew BLER LLS results to compare PA and non-PA strategies. Moreover, we chose the best PA, and utilized a SNR at 10% of the BLER in the SLS via EESM link-to-system mapping. Finally, we drew throughput and the outage performance to analyze the effectiveness of the power allocation strategy. From Figures 8 and 9, we validated the downlink power allocation strategy as working effectively in UAC networks.

**Author Contributions:** I.A. proposed the downlink power allocation strategy for UAC networks. He presented the approach to power offset using three kinds of pilot spacing and applied the PB concept to OFDM symbols for the UAC network. He drew the BLER curves from LLS and analyzed the SNR for PA and non-PA strategies. Finally, he chose the best PB for SLS and compared throughput and outage performance for PA and non-PA. K.C. was the technical leader for this manuscript. He suggested all the technical issues for the proposed downlink power allocation strategy for UAC networks and for the simulations. In addition, he corrected the simulation methodology in this manuscript and corrected mistakes in the simulation environment, as well as in the structure of the overall manuscript.

**Funding:** This research was supported as part of the project titled "Development of Distributed Underwater Monitoring and Control Networks", which was funded by the Ministry of Oceans and Fisheries, Korea.

**Conflicts of Interest:** The authors declare they have no conflicts of interest.

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
