# Peer review of "Downlink Power Allocation Strategy for Next-Generation Underwater Acoustic Communications Networks"

_electronics, doi:10.3390/electronics8111297_

Round 1

Reviewer 1 Report

This paper works on a downlink power allocation strategy for nect generation Underwater acoustic communications networks. The paper is written in a clear structure and provides thorough evaluation of the proposed strategy. Here are a couple suggestions. Section 2 and Section 3 can be expanded to help explain the topics. In section 2, authors state that in figure 2 the behavior of the UAC is quite similar to the statistical channel models in terrestrial scenarios. However, the paper only present the UAC. It's better to provide the terrestrial figures also or to provide comparisons/differences to show how similar it is. Section 3 can also be improved to describe the proposed strategy more clearly.

Author Response

Response Letter to Reviewer 1

Paper Title: Downlink Power Allocation Strategy for Next-Generation Underwater Acoustic Communications Networks

We make the necessary revision in the manuscript and rephrase them in the response letter. All the corrections are marked with blue in the manuscript. Authors would like to thank the reviewer for his careful revision which help to improve the quality of the manuscript.

Reviewer Number: 1

Reviewer's Report:

This paper works on a downlink power allocation strategy for next generation underwater acoustic communications networks. The paper is written in a clear structure and provides thorough evaluation of the proposed strategy. Here are a couple suggestions. Section 2 and Section 3 can be expanded to help explain the topics. In section 2, authors state that in figure 2 the behavior of the UAC is quite similar to the statistical channel models in terrestrial scenarios. However, the paper only present the UAC. It's better to provide the terrestrial figures also or to provide comparisons/differences to show how similar it is. Section 3 can also be improved to describe the proposed strategy more clearly.

Ans: Thanks for the comment. We expanded the Section 2 and Section 3 according to reviewer’s suggestion, by adding more explanations to clearly understand the topic. (Pages 4 to 7).

     We added the example behavior of terrestrial Winner II (scenario B1) channel model is shown in Fig. 3 [35] (Page 4) for Section 2. By comparing the Fig. 2 and Fig. 3, we can easily see that the behavior of the UAC south sea channel is quite similar to the statistical channel models in terrestrial scenarios (e.g., Winner II, etc.). Hence, the delay versus amplitude of discrete curves are showing the similar behavior for UAC south fading channel model and terrestrial Winner II fading channel model in Fig. 2 and Fig. 3, respectively. This details we also added in the manuscript (Section 2, Page 5).

     Moreover, in Section 3, we added more explanation to understand the DL PA strategy more clearly (Section 3, Pages 6, 7). In this section, we explain the proposed DL PA strategy for the UAC network by referring the 3rd generation partnership project (3GPP) long-term evolution (LTE) system DL PA strategy [36 to 38]. As there is no DL PA strategy is available in the UAC specification documents like 3GPP LTE, so it is necessary to refer the already well-established DL PA approaches of LTE for the UAC networks. The 3GPP has defined two parameters PA and PB which actually controls the downlink power allocation. PA represents the power of the reference signal as compared to OFDM symbol A and PB represent the power of the reference elements of OFDM symbol B as compared to OFDM symbol A. PA and PB have various combination of different values which can be used to utilize the maximum power of network, and as per our requirements.  

     The main objective is to avoid the power variations at the receiving side by maintaining the constant power using the DL PA strategy. The boost in the reference signal power is caused the compensation of the type B symbols (which contains the reference symbols) as compared to the type A symbols (that do not have the reference symbols). The data signal power always depends on the number of allocated resource blocks. This allocation of resource blocks can be changed sub-frame by sub-frame. It is insured that the overall OFDM symbol power should remain constant during incorporating the PA and PB.

Reviewer 2 Report

In this paper, the authors analyze a downlink power allocation strategy for next-generation UAC networks. Some suggestions are proposed as follows.

The authors are suggested to provide a comprehensive review for existing works. The authors are suggest to provide more simulation parameters to verify the simulation results.

Author Response

Response Letter to Reviewer 2

Paper Title: Downlink Power Allocation Strategy for Next-Generation Underwater Acoustic Communications Networks

We make the necessary revision in the manuscript and rephrase them in the response letter. All the corrections are marked with blue in the manuscript.

Authors would like to thank the reviewer for his careful revision which help to improve the quality of the manuscript.

Reviewer Number: 2

Reviewer's Report:

In this paper, the authors analyze a downlink power allocation strategy for next-generation UAC networks. Some suggestions are proposed as follows.

    The authors are suggested to provide a comprehensive review for existing works. The authors are suggest to provide more simulation parameters to verify the simulation results

Ans: Thanks for the comment. We added the comprehensive review of the existing works according to reviewer’s suggestion. Many researches have been proposed to assess the features of the UAC networks in the existing literature. However, enormous features related to UAC networks are still needed to be addressed on emergent basis such as considering the complicated scenarios like terrestrial cellular networks [14-27]. The existing works related to underwater communication have been considered simple networks architectures and mostly focused on the assessment of underwater channel model and routing protocols [28-31]. In order to fill this gap, we consider the complicated scenario like terrestrial networks and employ the downlink power allocation strategy for the UAC networks. However, many researchers have been worked on the UAC power allocation issues but they did not consider the complicated scenarios.

    Differences in System Methodologies in the Literature: In [32], equal transmission power control scheme is applied for the clustered based network approach for UAC network. The major difference of this work is employing the power allocation scheme for non-orthogonal multiple access while we utilize the OFDM technique for UAC networks. In [33], authors investigate the power allocation strategy for energy harvesting in the UAC networks. They considered the two scenario for knowing the channel state information and applied the stochastic dynamic programming to find the optimize power allocation UAC networks. The major different of this work is adopted the power allocation for energy harvesting in the UAC network. In [34], author jointly utilized the power and frequency allocation strategy to minimize the energy consumption for the UAC networks. The major difference of this work is selectin the proper center frequency, bandwidth and transmission using the routing protocols. Hence, the different approaches and system design in the existing works [32-34] resulted in different system parameters. Therefore, it is very difficult to compare the proposed DL PA strategy with the other researches. To the best of our knowledge, this work is the first to present the downlink power allocation issues using the power allocation strategy for the UAC networks. We have added this information in the manuscript (Section 1, Pages 1, 2).

    Moreover, we added more simulation parameters according to reviewer’s suggestion in order to verify the simulation results in the manuscript. (Section 4, Page 8).

Reviewer 3 Report

Main problem for the reader: It is unclear how the authors analyze the downlink power allocation strategy for  the UAC networks, how many nodes inside the scenario, collisions with different power allocation transmissions. The results in figure 4 to 9 cannot be repeated by others.

Please use a common network simulator, e.g. DESERT under ns2

http://telecom.dei.unipd.it/ns/desert/DESERT2_HTML_doxygen_doc/index.html

to repeat your work. Science has to be repeated by others. Please describe the scenario

and LUT tables.

some other points: line 42 swap to next page / more description for figure 2 is needed

Author Response

Response Letter to Reviewer 3

Paper Title: Downlink Power Allocation Strategy for Next-Generation Underwater Acoustic Communications Networks

We make the necessary revision in the manuscript and rephrase them in the response letter. All the corrections are marked with blue in the manuscript. Authors would like to thank the reviewer for his careful revision which help to improve the quality of the manuscript.

Reviewer Number: 3

Reviewer's Report:

The Main problem for the reader: It is unclear how the authors analyze the downlink power allocation strategy for the UAC networks, how many nodes inside the scenario, collisions with different power allocation transmissions. The results in figure 4 to 9 cannot be repeated by others.

Please use a common network simulator, e.g. DESERT under ns2

http://telecom.dei.unipd.it/ns/desert/DESERT2_HTML_doxygen_doc/index.html

to repeat your work. Science has to be repeated by others. Please describe the scenario

and LUT tables.

some other points: line 42 swap to next page / more description for figure 2 is needed

Ans: Thanks for the Comment. We added more information about the proposed downlink power allocation strategy and also explain more about the considered scenario according to according to reviewer’s suggestion.

    The scenario of acoustic communication between the UBSC and UBS is quite similar to the terrestrial cellular communication such as base stations and users, respectively. Therefore, the UBSs exist in the region of interest can be considered as the only users or receivers where the scheduling, outage, and throughput calculation is performed while the transmitters and receivers in 1st tier are considered as the interference providing nodes. Hence, the downlink power allocation strategy is implemented based on the mentioned scenario of Fig. 1.

    We built Matlab-based LLS and SLS platforms, and employ the DL power allocation strategy for the next-generation UAC networks by referring the terrestrial cellular network communication approaches [6-8]. This work is continuation of our previous work in [15] where we analyzed the effective SNR mapping and link adaptation strategies for UAC networks. So, that is the main reason we did not utilize the common network simulators, for example DESERT under NS2. We added this information in the manuscript (Section 2, Page 4).

    Moreover, in Section 3, we added more explanation to understand the DL PA strategy more clearly (Section 3, Pages 6, 7). In this section, we explain the proposed DL PA strategy for the UAC network by referring the 3rd generation partnership project (3GPP) long-term evolution (LTE) system DL PA strategy [36 to 38]. As there is no DL PA strategy is available in the UAC specification documents like 3GPP LTE, so it is necessary to refer the already well-established DL PA approaches of LTE for the UAC networks. The 3GPP has defined two parameters PA and PB which actually controls the downlink power allocation. PA represents the power of the reference signal as compared to OFDM symbol A and PB represent the power of the reference elements of OFDM symbol B as compared to OFDM symbol A. PA and PB have various combination of different values which can be used to utilize the maximum power of network, and as per our requirements.  

    The main objective is to avoid the power variations at the receiving side by maintaining the constant power using the DL PA strategy. The boost in the reference signal power is caused the compensation of the type B symbols (which contains the reference symbols) as compared to the type A symbols (that do not have the reference symbols). The data signal power always depends on the number of allocated resource blocks. This allocation of resource blocks can be changed sub-frame by sub-frame. It is insured that the overall OFDM symbol power should remain constant during incorporating the PA and PB.

    Furthermore, we added more information about the scenario and added the look-up table in the manuscript according to reviewer’s suggestion (Section 2, Page 5). At LLS, we draw the BLER curves by considering the useful modulation and coding schemes levels [15], as shown in Fig. 4. The corresponding look-up table is listed in Table I. We take the SNR values at 10 percent of BLER curves for the nine modulation and coding scheme levels and utilize these values in SLS for assessing the DL power allocation strategy.

Round 2

Reviewer 3 Report

Thanks for your view point